# Digital Irrigated Agriculture: Towards a Framework for Comprehensive Analysis of Decision Processes under Uncertainty

**Francesco Cavazza [1],\*[ID], Francesco Galioto [1][ID], Meri Raggi [2][ID] and Davide Viaggi [1][ID]**

[1] Department of Agricultural and Food Sciences, University of Bologna, Viale Fanin 50, 40127 Bologna, Italy; francesco.galioto@unibo.it (F.G.); davide.viaggi@unibo.it (D.V.)

[2] Department of Statistical Sciences, University of Bologna, Via delle Belle Arti 41, 40126 Bologna, Italy; meri.raggi@unibo.it

\* Correspondence: francesco.cavazza7@unibo.it

**Abstract:** Several studies address the topic of Information and Communication Technologies (ICT) adoption in irrigated agriculture. Many of these studies testify on the growing importance of ICT in influencing the evolution of the sector, especially by bringing down information barriers. While the potentialities of such technologies are widely investigated and confirmed, there is still a gap in understanding and modeling decisions on ICT information implementation. This gap concerns, in particular, accounting for all the aspects of uncertainty which are mainly due to a lack of knowledge on the reliability of ICT and on the errors of ICT-information. Overall, such uncertainties might affect Decision Makers' (DM's) behavior and hamper ICT uptake. To support policy makers in the designing of uncertainty-management policies for the achievement of the benefits of a digital irrigated agriculture, we further investigated the topic of uncertainty modelling in ICT uptake decisions. To do so, we reviewed the economic literature on ambiguity, in the context of the wider literature on decision making under uncertainty in order to explore its potential for better modeling ICT uptake decisions. Findings from the discussed literature confirm the capabilities of this approach to yield a deeper understanding of decision processes when the reliability of ICT is unknown and provides better insights on how behavioral barriers to the achievement of potential ICT-benefits can be overcome. Policy implications to accompany the sector in the digitalization process include mainly: (a) defining new approaches for ICT-developers to tailor platforms to answer heterogeneous DMs' needs; (b) establish uncertainty-management policies complementary to DM tools adoption.

**Keywords:** information and communication technologies; digital irrigated agriculture; artificial intelligence; ambiguity; risk

## 1. Introduction and Objectives

Information and Communication Technology (ICT) is defined by the World Bank as "( … ) any device, tool, or application that permits the exchange or collection of data through interaction or transmission" [1]. As in everyday life we have seen the proliferation of ICTs contributing to support many decisions, also agriculture is taking part to this digitalization process. In this sector, ICTs mainly include Decision Support Systems (DSS), tools based on Artificial Intelligence (AI), Internet of Things (IoT), Climate Services, Geographic Information Systems (GIS) and many other digital tools which offer Decision Makers (DMs) a wide variety of support [2].

Because of the great potential of ICTs, some authors call this phenomena the Digital Agriculture Revolution (DAR), and they believe it can help to solve some challenges the sector is facing [3]. This expression highlights just how important the stage of agricultural development we are living in according to scholars. While the DAR can be expected to be comparable in magnitude with the "green revolution" [1,3], the innovation paradigm brought by the DAR is different. During the green revolution technologies were aimed at altering the agroecosystem through fertilizers, pesticides and genetics. The DAR is altering the decision environment through information provision. Accordingly, all digital technologies in agriculture have one common element, which is in the use and generation of data. With it, the new platforms generate information aimed at supporting decisions by lowering uncertainty. Decision processes can now move from precautionary and inefficient choices forced by uncertainty, to decisions based on sound information. In this context, irrigation and water management are one of the key sectors where ICT-information would have the most important applications with the highest benefits [2,4]. Many ICT have been developed in irrigation and water management [4–6]. This sector is more susceptible to uncertainties than others and ICT have a great potential to lower uncertainty and help facing the challenges posed by conflicting uses, water scarcity and extreme weather events [6–9].

In the industrial and utility sectors, the adoption of ICT to support decisions on water use and allocation is rapidly growing in what is identified by the International Water Association as the *digital water journey* [10]. In irrigated agriculture, the digital water journey is more difficult due to the intrinsic characteristics of the sector. Here, dynamics for ICT implementation are extremely complex and lack of infrastructures, lack of supporting technologies, financial restrictions and knowledge gaps often pose significant barriers to ICT diffusion [7,11]. More trivially, in some occasions Water Authorities (WAs) and farmers might decide to not implement an ICT because the information conveyed by the ICT is not mature enough or not suitable for the purpose for which it is used [12]. For example, the low accuracy of available devices makes many platforms useless to aid farmers' irrigation decisions [8]. The low accuracy of the information provided through existing ICT devices is also documented at the level of WAs for the allocation of water resources through open-air canals [7]. Even when the information conveyed through ICT is mature enough to generate appreciable benefits, behavioral barriers can hinder the digitalization process [13]. As a result, the digital transition for irrigation management cannot be self-accomplished by the sector. Constraints to digitalization will not only slow ICT implementation, but they also risk compromising the progress of ICT innovations in agriculture, leaving the sector with obsolete tools and not effective to face issues of water scarcity and conflicting uses [10]. These conditions highlight the need to understand decision processes and to design policies to favor ICT development and uptake for water management in agriculture [14,15].

In the applied economics literature, there are several studies addressing the topic of ICT implementation in agriculture and water management [4,5,16]. The most important works in applied economics estimate the benefits of ICT implementation by defining the circumstances in which information has a value for a DM [17]. Although scholars agree on the theoretical settings in which ICT-information is valuable, empirical applications show discordances and ICT-benefits are still unclear. Further, there are gaps in the modelling of decision on ICT implementation to account for the uncertainty settings which affect DM's behavior and might impede ICT uptake. Accordingly, there are different sources of uncertainty around the decision for ICT implementation. These can be caused by the following issues: (i) the ICT provides imperfect information (information is not 100% reliable) and (ii) the DM does not know ICT's reliability (the ICT is new to the DM). However, uncertainty is often modeled not distinguishing between its different sources and issues generated by a lack of knowledge on ICT reliability are overlooked. This does not allow to understand how perceptions on ICT reliability affect the farmer's or Water Authority's (WA) behavior and, in turn, the decision on ICT implementation.

In this paper, we reviewed the economic literature on decision making under uncertainty to seek for the theoretical basis for better modelling ICT uptake decisions in irrigated agriculture. Between theories, we found the one of ambiguity developed by Ellsberg (1961) to be suited to help modelling the uncertainty settings affecting ICT uptake for irrigation management. In particular, we represent ambiguity as the share of uncertainty generated by not knowing ICT's reliability and risk as the error in ICT-information. This distinction will yield a deeper understanding of decision processes when the reliability of ICT is not known. Further, it allows to model the process of familiarity which occurs as the DM gains experience on the ICT. As familiarity can be a powerful tool to ease ICT-uptake we will also consider policy implications to favor the achievement of the digital irrigated agriculture

The novelty of this research is in the application of economic literature's theories to provide the basis needed to model the uncertainty settings around ICT adoption for irrigation management. In most of applied studies on ICT adoption in agriculture risk is modeled in the decision process to be the only element which shapes uncertainty. Contrarily to these studies, we introduce the concept of ambiguity explaining the share of uncertainty rising when the reliability of a new ICT is unknown.

The remainder of this paper is organized as follows: in the next section we will frame the problems and gaps by reviewing literature on ICT implementation and water management. In Section 2, we will analyze the most relevant theories and applications of decision making under uncertainty. In Section 3, we will focus on the theory developed by Ellsberg [18] and apply it to the context of our research; finally, in Sections 4 and 5 we respectively discuss what we have learned from this review and draw conclusions and policy implications.

## 2. Trends on ICT Adoption

The topic of ICT adoption in agriculture is of growing relevance and numerous ICT development initiatives have been carried out to aid the sector [19]. The use of such technologies for water/irrigation management is considered one of the most promising applications, although their application is still predominantly pioneering. This might be confirmed by the growing body of articles published on the topic (Figure 1) and by the interest raised in the applied economic literature [4,15,20].

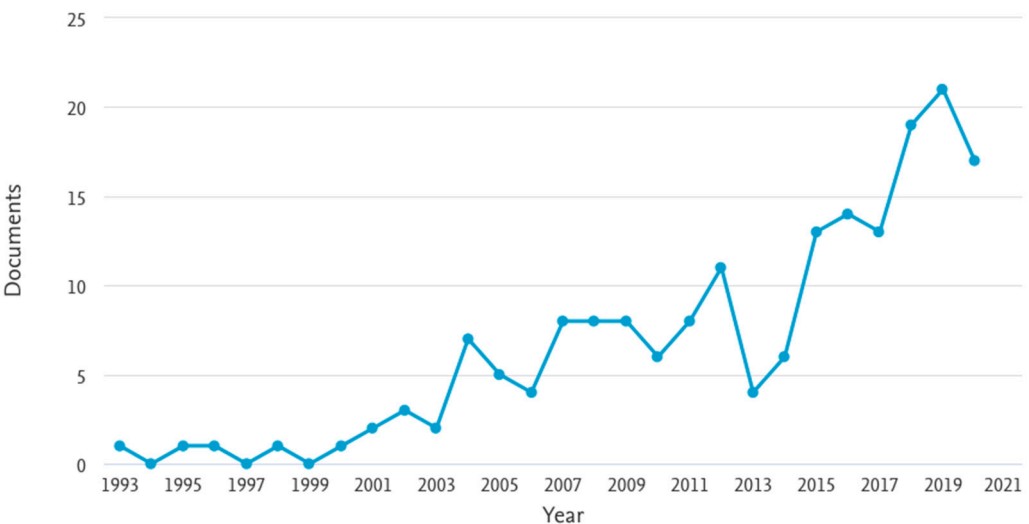

**Figure 1.** N. of published articles on Information and Communication Technologies (ICT) adoption in agriculture water management. Source: own elaboration from data obtained with the search in Scopus (dated 30/08/2020) having the following parameters: TITLE-ABS-KEY ((ICT OR DSS OR "climate services") AND agriculture AND ("water management" OR irrigation)).

Notwithstanding the great interest shown by scientists for ICT, in many cases their impact is not well-defined and results of applied economic studies are extremely variable (Aker et al., 2016). ICT can provide benefits only if the delivered information is eventually implemented by DMs to improve decisions [21]. Apparently, this occurs only when the information content answers DMs' needs [22] and when the way in which information is conveyed such as timeliness of information provision or spatial scale allows the DM to implement it [21]. It is not only information usability which conditions ICT-benefits; DM's behavior can strongly affect ICT-information implementation too. Tumbo et al. [23] in their analyses found that farmers in Tanzania are seeking ICT-information to adapt to CC, especially in their irrigation activities. Though, they highlight that uncertainty on ICT reliability limits information uptake. Nesheim et al. [24] found that the use of ICT in India has not reached its potential and many farmers do not implement the forecast received. This is mainly due to farmers not understanding information received or having doubts on ICT reliability [24]. Kirchoff et al. [13] carried out qualitative and quantitative analyses on ICT adoption for water management in the U. S. and Brazil. They found that, when DMs perceived information reliable, this helped information uptake and efficient water management; while, in case of skepticism, information was not implemented [13]. Hawoth et al. [25] carried out a review on ICT development initiatives across the world. Their findings underline that between the 27 ICT only few were used by the addressed farmers. They identify a number of barriers that opposes to ICT-information implementation, among which the most relevant ones are associated with information quality and perceived reliability. O'Mahony et al. [9] showed that lack of knowledge on the technology's reliability can even prevent achieving sustainable water management practices in Australian agriculture. It is not surprising that if the ICT or the information itself are considered unreliable or are mis-interpreted, DMs do not implement the message received and gain no benefit from the technology [19].

Such findings are confirmed by several quantitative studies, out of which a selection of empirical application for ICT benefits estimations is reported in Table 1. The table highlights that almost all studies testify that benefits can be null, often because of the presence of barriers related to the decision environment or to the information itself. Although the table is not comprehensive of all the studies done in the field, it is enough to testify the inconsistency of the impacts brought by the implementation of existing ICT in agriculture. To gain better insights from this topic, please refer to the works of Jeuland et al. [4] and Meza et al. [16] which provide respectively reviews of studies for ICT-information benefits estimation in water management (predominantly irrigation) and in the broader agricultural sector.

In summary, some of the discussed studies highlight that not all the DMs who receive information actually implement it, and both technical, but most of all, behavioral barriers undermine information implementation. Further, even when ICT information is implemented and put into action, quantitative studies highlight that in many occasions there is a lack of benefits from ICT-aided decision processes. This poses serious limits for ICT development in agriculture because investments are not easily justifiable and returns doubtful [4].

**Table 1.** Examples of estimation of benefits from information implementation.

| Authors | Journal | Field of Application | Case Study Location | Type of Information | Benefits | Barriers |
|---------|---------|---------------------|---------------------|-------------------|----------|----------|
| [26] | Journal of Environmental Management | Water quality | Netherlands | Earth observations | 0–0.4 mln€ | Quality of information; uncertainty on reliability; DM's prior expectations; lack of confidence on informed actions; political barriers |
| [27] | Regional Environmental Change | Agriculture | Southeastern United States | Climate information | 0–3.4 mln$ | Agricultural sector; region; use of discrete-type forecasts |
| [28] | Australian Journal of Agricultural and Resource Economics | Agriculture | Australia | Seasonal forecasts | 0–55 $/ha | Quality of information; crop-planting time; poor timeliness of information provision |
| [29] | Agricultural Systems | Crop plan | U.S.A. | Weather forecasts | 0–12 $/ha | Quality of information; management strategies; uncertainty on reliability; crops revenues |
| [30] | Climatic Change | Agriculture | U.S.A. | Decadal climate variability | 0–1.7 $/ha | Quality of information; cropmix; irrigation |
| [8] | Water | Irrigation | Denmark, Portugal, Spain, Greece Italy | Crop water requirements | 0–250 €/ha | Quality of information; irrigation water cost; output price; risk aversion |
| [7] | Water | Water management in agriculture | Italy | Long- and short-term crop water requirements | 0–26.2 €/ha | Quality of information; the stake in the decision process; time of information provision; land use; water delivery system |
| [31] | European Journal of Agronomy | Sugarcane irrigation | Australia | Seasonal forecasts | 0–200 $/ha | Quality of information; uncertainty on reliability; type of forecast |
| [32] | Agricultural water management | Cotton | China | Weather forecasts and crop stress | 600–2.000 $/ha | Irrigation methods and scheduling |

## 3. Behavioral Approaches to Uncertainty

### 3.1. Perceptions and Attitudes

Decisions are affected by the DM's behavioral perspective with a relevant role played by attitudes toward uncertainty [33]. In every decision under uncertainty, risk is involved and aversion to it might cause DMs to sacrifice part of their revenues to lower the variability of uncertain outcomes [16]. This is confirmed by [31] for the irrigating sector, where uncertainty leads to the implementation of inefficient precautionary actions. Information provision could lower the unpredicted climate variability and allows uncertainty averse DMs to rationalize protective actions and increase their expected utility. However, this is not always true and, depending on the source, a new piece of information could even raise outcomes variability under the DM's perspective [34]. The topic of information implementation and aversion to uncertainty is widely debated in literature. For example, [35] found that uncertainty aversion raises utility from information only when the information provided is perfect. A relevant contribution to the topic is given by [36] who showed a series of scenarios where uncertainty averse DMs did or did not seek for a new piece of imperfect information. They conclude that the relation between benefits from receiving new pieces of information and the degree of aversion toward uncertainty is very situation dependent. In particular, DMs who have extreme attitudes toward uncertainty (both very risk averse and risk seeker) do not seek for information when they have to decide whether to implement a new technology whose reliability is unknown.

As highlighted in the PhD dissertation by Cavazza [37] and in the paper by Nesheim et al. [24], besides attitudes toward uncertainty, the perception of uncertainty over a phenomenon is extremely important in affecting decisions for ICT adoption. Further, in case of information provision, also the perceived uncertainty over a new piece of information describing the same phenomenon is found to be limiting information uptake [31]. One strategy proposed to solve this uncertainty is in the estimation of the forecast reliability and in the incorporation of this information in the message itself [29]. This is commonly done with probabilistic climate forecasts, but still DMs could doubt on the probability estimation in itself. This explains one of the reasons why DM's beliefs are needed to be accounted for in decision analyses [38]. Nevertheless, in literature important challenges remain in understanding the role of perceptions and attitudes [4] and most of the studies in applied economics make relevant behavioral assumptions to overcome the issue [39]. This gap in applied economics motivated us to seek for theories in the broader economic literature to build the theoretical basis needed to develop research and better understand decision processes of ICT implementation under uncertainty.

### 3.2. Toward a Framework for Uncertainty Modelling

To solve the relation between uncertainty aversion, perceptions and information implementation, a better picture of what uncertainty is built of is needed. Useful insights can be gained from decision theories developed in the economic literature and describing decision making under uncertainty.

The representation of preferences over uncertain actions, meaning actions whose payoff is dependent on the emergence of uncertain states, is firstly addressed by von Neumann and Morgenstern [40]. They developed the Bernoullian [41] concept of expected utility and outlined the dominant theory to describe attitudes toward uncertainty [38]. The theory is defined on the basis of the following four axioms:

- A1—Completeness: every state of the world involved in a decision can be completely ranked.
- A2—Transitivity: the property of transitivity holds for preferences for alternative states of the world.
- A3—Independence: preferences for alternative states are context-independent.
- A4—Continuity: preferences for alternative states are expressed on a nominal scale.

By holding these four axioms, a wide variety of utility functions have been developed in literature to represent DM's preferences under uncertainty [42]. The common characteristic of all

von Neuman–Morgenstern's utility functions is that concave functions represent an uncertainty aversion behavior. While, the function represents neutrality to uncertainty if it is linear, and propensity to uncertainty is modeled if it is convex.

With regards to the representation of perceptions over uncertainty, the first theory developed in the economic literature is the one proposed by [43]. He was the earliest who gave a complete picture of uncertainty. His main contribution was in distinguishing between: (i) measurable uncertainty, occurring when the statistical frequencies of events are known to the DM and (ii) un-measurable uncertainty when they are not. The former uncertainty was defined as pure risk, while the concept of un-measurable uncertainty remained unclear [44]. Even Knight in his work considered the fact that under un-measurable uncertainty DMs formed subjective "probability estimates" and treated them as risk. Accordingly, he postulated that "there is no difference for conduct between a measurable risk and an unmeasurable uncertainty". The idea of subjective probabilities was further developed by Keynes [45] and Ramsey [46] who both contributed to the formulation of the concept of "degrees of belief" representing the DMs' rational probabilistic estimation in condition of un-measurable uncertainties. Because of the understanding of the former authors and the vast literature following, the concept of subjective probabilities had a large success in recognizing that most of the decisions were characterized by the absence of measurable frequencies. This is especially true after Savage [47] built on it the theory of Subjective Expected Utility (SEU). Savage defined a theory of decision making under uncertainty characterized by preferences over acts with uncertain outcomes being compliant of 7 axioms, out of which the most studied ones are described as follows:

- A1—Complete ordering: there is a preference relation over uncertain actions which is complete, reflexive and transitive.
- A2—Sure-Thing principle: the preference relation over two uncertain actions is not affected by their payoffs in states where both actions have the same payoff.
- A3—State-wise monotonicity: in a given state, one action is preferred to another if and only if their payoff is equally ordered.
- A4—Independence between payoffs and probabilities: given preferences between payoffs, the choice between two uncertain actions is not affected by the value of the payoffs.

When representing choices under uncertainty, the SEU model developed by Savage allowed to distinguish between subjective probability and preferences. Often, these take the form of a von Neumann–Morgenstern's [40] utility function. The practical implication of the theory is that the DM builds probabilistic representations of states and uses these linearly by weighting uncertain payoffs. With the application of the theory, observed decisions under uncertainty can be used to assess DM's beliefs [44]. This is true even in presence of measured frequencies where the DM might doubt their reliability and assuming that they will represent future likelihoods is a subjective judgment per se [38].

Despite the great success of SEU, there have been applications that showed some of its limitations. Between these, Allais [48] highlighted that, even with objective lotteries, preferences are context-dependent and, in some cases, the von Neuman–Morgenstern's independence axiom did not hold. In turn, Ellsberg [18] showed exceptions of Savage postulates in describing perceptions. Finally, Kahneman [49], by developing prospect theory, presented that often both preferences and perceptions did not follow von Neuman–Morgenstern's and Savage's axioms respectively. Other theories describing perceptions and preferences have been developed in literature, but the one considered in this section are the ones mostly used in the economic literature [44].

## 4. Ambiguity and ICT-Information Implementation

### 4.1. The Theory of Ambiguity and Ambiguity Aversion

As highlighted in the previous section, many theories have been developed in literature to explain decision making under uncertainty. Between these, most of the studies consider risk to be

the only element of uncertainty in decisions. However, this does not allow to explain some situations as those described by Ellsberg [18]. In his work, Ellsberg focused on un-measurable uncertainties to show important exceptions to the Savage axioms and in particular to the independence axiom. These exceptions impaired the capability of SEU to represent decisions and to highlighted that it is not only risk, but also ambiguity driving decisions. He considered a series of thought experiments, out of which, the most famous is the one involving two urns. There is a transparent urn with observable content of 50 red and 50 black balls and another opaque urn with the same amount of balls, but with unknown ratio between the two colors. While in the transparent urn the DM faces a situation of pure risk, because probabilities are observable and measurable; the bet in the opaque urn is different because such measurement is not possible. To describe the opaque urn's content (hence the probability of a specific ball's color) hundred combinations between red and black balls are possible. This raises uncertainty over which combination is the one really describing the urn's content. In such settings, Ellsberg considered that most of DMs would have preferred to place a bet on the ball's color from a draw in the transparent urn, instead of a draw from the urn with unknown balls' ratio. For a given prize in the bet, he proved that this choice cannot only be driven by a mere difference in subjective probabilities. Rather, the preference to bet on the transparent urn and, in general, the preference for gambles with known probabilities, is driven by a behavioral phenomenon called Ambiguity Aversion (AA). The concept of AA is similar to Risk Aversion (RA), where ambiguity is identified as:

> "the nature of one's information concerning the relative likelihood of events a quality depending on the amount, type, reliability and 'unanimity' of information, and giving rise to one's degree of 'confidence' in an estimation of relative likelihoods." [18].

Or, in the words of Camerer and Weber [50], ambiguity is more clearly defined as "uncertainty about probability, created by missing information that is relevant and could be known". Overall, similarly to what Knight did, uncertainty is then characterized by two elements: (i) risk, represented by the share of measurable uncertainty or estimated trough subjective probabilities, and (ii) ambiguity, expressing the degree of confidence over these probability estimations.

Because ambiguity affects a large share of decisions under uncertainty, there have been several experimental studies showing its relevance in decision making. Results highlight that AA impacts are comparable with, if not higher than, RA [51]. Further, these studies showed that AA is the main preference behavior of DMs under ambiguity, because DMs dislike situations where more than one probability estimation is possible [52]. The presence of AA implies that decisions do not only reveal subjective probabilities, but also relative preferences for expected outcomes. There are situations, such as in the urns' examples (and, as described later, in ICT-information implementation), in which we need to distinguish between the two elements of uncertainty to understand DM's behavior. In these situations, SEU models cannot be applied in a straightforward way but ambiguity-sensitive preferences have to be accounted for. Further, the uncertainty framing proposed allows to model the process of familiarity which occurs as the DM gains experience with a new phenomenon or a new technology, such in the case of ICT.

## 4.2. Ambiguity and ICT-Information

The capability of the theory developed by Ellsberg to provide a complete picture of the elements shaping uncertainty explains its large adoption in different decision problems [44]. In the majority of decisions, DMs have to cope with a certain level of risk and ambiguity. The latter expresses the degree of confidence the DM puts on the former [53]. This situation is typical with decisions under climate change [54]. Here, probabilistic distributions built with past records are mistrusted to be representative of future climate trends and different models propose different projections. Ambiguity can be found in climate forecasts too. Uncertainty in forecasts is characterized by two elements: (i) the intrinsic variability of climate events, represented by probability distributions and (ii) the uncertainty about the forecast itself [55]. The first uncertainty can be expressed as risk, over which, the second uncertainty

emerges because the DM does not know whether the forecast is reliable. This lack of knowledge raise ambiguity because several forecasts could be delivered and the DM is not sure whether the one received is really representing future states. One common way to deal with such complex uncertainty settings in forecasts is through uncertainty folding [56]. It consists in folding ambiguity with risk to obtain a single probability estimate used as input in SEU models. The method is rather simple and appealing. However, it has been shown by Allen and Eckel [56] that it fails in representing real decision making, due to important informational losses.

As with forecasts information, ambiguity rises even with the adoption of new technologies [57]. When a DM is faced with a new technology, he is uncertain on the probability of such technology to be good performing. In this context, even if the DM has expectations or information from the developer, neither of the two probability estimations can be reliably assessed. As a result, in the first stages of a new technology, AA has the potential to strongly limit its diffusion. Only after having gained enough experience with the new technology, prior expectations can be confirmed or rejected and ambiguity solved in a learning process [58]. Most of the applied economic literature considers ambiguity generated by new technologies to be risk, thereby losing precious DM's behavioral insights [59]. If risk and ambiguity are treated as one, risk-averse and ambiguity-averse DMs would behave equally. Both would be less prone in experimenting new technologies or implementing new forecast information, but this does not always happen. A new technology could be risk-reducing but ambiguous, or the opposite, it is possible to gain information reducing ambiguity but not risk [60]. In this complex framework, Snow [61] defined the relation between information value and AA, where information reducing ambiguity is always sought by AA individuals, and the benefits from its implementation rise with the degree of aversion. The same applies with information reducing risk and RA. However, while information solving ambiguity is valued only by AA DMs, if information solving risk completely disclose states of the world, its benefits rise with both AA and RA [61]. These phenomena are extremely important in the context of new technology adoption. If a new technology reduces the variability of outcomes, it is risk-decreasing but, as said before, it might raise ambiguity. Here, RA plays in favor of the technology, while AA might limit its implementation.

## 5. Discussion

### 5.1. Lessons Learned from Literature

In the literature, there are several studies addressing the topic of ICT implementation in agriculture and water management [4,16]. Nevertheless, results are contradictory, and none provides a comprehensive assessment. Despite many works underline how aversion to uncertainty can compromise ICT implementation and ICT-benefits, the uncertainty settings around the decision environment are seldomly thoroughly tackled. To fill this gap and to provide the needed theoretical foundation for new decision models, we analyzed how uncertainty is represented along the economic literature's history. Between the theories developed to better understand the various behavioral aspects of decision making under uncertainty, the one of AA is considered the most useful to understand the behavior of forecast implementation [56] or new technology adoption [58]. Innovative technologies are frequently raising ambiguity either because the probability of a good performance is uncertain or, as in the case of ICT-climate-information, because they provide probabilistic information whose reliability is unknown. A new platform developed to deliver information to DMs can be considered risk reducing because it lowers variance in the upcoming states' distribution thanks to a better knowledge on phenomena. If so, it is always positively valued and implemented by RA DMs, given technical barriers are overcame. All the same, the DM does not know if the platform (hence the piece of information received) is reliable. This might raise ambiguity and is discouraging AA individuals to implement ICT-information until they do not gain enough information, or even better experience, on the technology's reliability. Accordingly, with experience the DM would learn if the technology is reliable and solve ambiguity [62]. The phenomenon is identified as familiarity with a technology and

might allow AA individuals to implement information received. While risk is often intrinsic to the technology and can hardly be modified, by providing ambiguity-reducing information or allowing DMs experience with the ICT, even AA individuals might find benefits from information.

The approach proposed to deal with uncertainty in ICT adoption is expected to be capable of providing the required framing for applied models aiming at further deepening the issue of low ICT-information implementation. Research is suggested on the topic, especially focusing on modelling application of ambiguity sensitive preferences. In this field, theoretical alternatives are proposed [44]. Between these, the smooth ambiguity model developed by Klibanoff, Marinacci and Mukerji [63] is considered the best performing in accounting for AA and RA behaviors [44]. It allows the separation between perceptions and attitudes, both with reference to risk and ambiguity. This would permit comparing the condition of ICT-information implementation when the DM is uncertain on its reliability and after he has gained enough experience on the technology's reliability. In such settings, uncertainty will be firstly made by risk and ambiguity, then, with experience, ambiguity vanishes, and risk remains unaltered.

*5.2. Limitations and Future Research*

Uncertainty on its own is unclear and its structure and impacts are often debated [44]. This is highlighted by the numerous theories developed in the economic literature, often one theory in contrast with the other. In this research we provided a representation of uncertainty where there is a clear separation between risk, which is exogenous and captured with the accuracy of the ICT, and ambiguity which is a characteristic of the DM's subjective belief. This is a simplification of real decision processes, where uncertainty is not dichotomic and its multiple forms might be indistinctly perceived by DMs. Here, the theory of ambiguity and AA was chosen by the authors because it allows to frame the uncertainty settings occurring when DMs have to decide whether to implement a new ICT or not. However, the capability of the uncertainty framing proposed has to be tested. This might cause over-estimations of the impacts that uncertainty-aversion has on the decisions. On the one hand, other theories might better explain reality and further research carrying out comparative experimental tests would be useful to assess the theoretical background most suitable to represent behaviors toward ICTs. On the other hand, the theory of AA is found by the authors to be the only which allows the modelling of decision dynamics on ICT implementation occurring along the process of familiarity. Because we believe familiarity to be a key target for policies, we consider the theory of AA the most useful in representing the uncertainty settings of the study. Accordingly, the uncertainty which is risk is hard to be modified because intrinsic to the decision environment. For example, risk can be due to: (i) the accuracy of the ICT, which in the short term cannot be raised more than what the state-of-the-art model offers; (ii) or the variability of climate events, which are exogenous to the decisions. In direct contrast, ambiguity can be lowered in the short term by allowing DMs experiencing with the ICT and building knowledge on the ICT reliability. At this end, policies have the capability to ease the process of familiarity, speeding up information uptake.

Despite the above described limitations, the uncertainty framing proposed in this paper allowed to highlight the main behavioral issues which hinder ICT implementation. Moreover, it can be extended and applied to the broader implementation of technologies in precision agriculture and where the digitalization process is taking place. Here, the use of new tools is widespread, but uncertainties on the real performances or reliability still affects implementation decisions. We consider particularly relevant the topics of: (i) result-based payments to compensate farmers for the provision of ecosystem services and (ii) weather-indexed insurances developed on the basis of agrometeorological models to provide new opportunities for risk management. Environmental uncertainties cause information asymmetries between farmers and the regulator, in case of result-based payments [64], or between farmers and the insurance company in case of weather indexed insurances [65]. ICT-information provision would have a potential key role in lowering uncertainties, while allowing the parts to be to be in the same informational settings. This is true for both result-based payment, where platforms

can disseminate to the supplier and the regulator information on the current ecosystem status [66], and index-insurance, where ICTs allow to increase transparency and real time monitoring of losses [67]. Nevertheless, the parts involved in the development of these tools might be unwilling to implement ICTs as they benefit from such asymmetries. Further, if lack of knowledge on the platform's reliability occurs, there will be differences in ambiguity perceptions. The impact of asymmetries in perceptions can be expected to lower the economic efficiency of these new insurance or policy tools.

## 6. Conclusions and Policy Implications

In this paper we started acknowledging the potentials of digital irrigated agriculture. At this end ICTs are recognized to be one of the most promising tools to aid the sector [68]. Through a brief review of applied economic studies, we highlighted a lack of success of many ICT-development initiatives [21]. Technical barriers undermine information implementation for irrigation management and benefits from ICT-aided decision processes are unclear and extremely variable. This poses constraints for ICT development and growth because investments are not easily justifiable and returns doubtful [4].

Other than technical barriers, one of the major issues highlighted in both qualitative and quantitative studies is that new ICT platforms generate uncertainty on information reliability. Because of this, DMs' behavior appears to be strongly limiting ICT-information implementation, but the topic remains to be deepened. Accordingly, we found no study in the applied economic literature to be addressing this issue and providing the needed theoretical support to model behaviors in ICT-aided decisions. To face this problem, we sought for support in the wider economic literature. Here, many theories have been developed, but the one considered to best fit with our uncertainty settings is the one of AA developed by Ellsberg [18]. With it, we framed uncertainty and explained why ambiguity and AA are key elements in describing the issue of low rates in new ICT-information implementation.

Our proposed approach to address uncertainty in ICT implementation is different from most of applied studies on ICT adoption in agriculture. Here, RA is often considered to be the only behavioral driver for technology adoption and different sources of uncertainty are treated indistinctly. Because ICT are peculiar technologies providing information capable of reducing risk but raising ambiguity, AA needs to be analyzed to better explain the process of ICT-information implementation. Accordingly, since the reliability of new platforms is uncertain, an AA behavior can impede DMs in implementing ICT-information. Key role is played by a DM experiencing with the new technology without necessarily need to buy information received or put information into actions at his own risk. This would help WAs or farmers to learn whether the ICT provided is reliable or not, therefore reducing or solving ambiguity.

What we have learned on how ambiguity enters into ICT-information uptake in irrigated agriculture is of strong policy relevance too. Accordingly, to be able to implement optimal strategies for ICT development and adoption, it is important to understand how DMs perceive new pieces of information [53]. To act in favor of the digital irrigated agriculture, there could be three main categories of policies developed: (1) ICT-development policies; (2) uncertainty-management policies and (3) agricultural and water policies. These are motivated by the potential economic benefits from ICT-aided irrigation plans and should favor an efficient digitalization process. In particular, the policy implications we drew are:

(1) ICT-development policies are needed to overcome issues of information usability and boost ICT-potentials. The simple information provision is not sufficient to allow its implementation because of local specificities in the end-user's information environment. This is especially true for irrigation management, where climate and technical elements can vary significantly between decision contexts and can hinder ICT-information implementation. At this end, ICT are suggested to aim at delivering information tailored to farmers' or WAs' specific needs [22]. However, most of ICT projects are characterized by a top down technological development where platforms are designed without involving end-users [3]. This causes phenomena such as the "loading dock" [69] where end-users are provided with relevant climate information which has no use in reality because its form is incompatible with actual decision making [21]. Further, this approach

feeds skepticism toward ICT reliability, when DMs have never experienced the new platform [3]. Rather than a top down approach to ICT development, a bottom up involvement of farmers and WAs is suggested. If ICT developers gathered more information and feedbacks from end-users, they would be better assured that barriers to information usability are overcame. However, a participatory ICT development process is likely to be more complex, to be longer and to incur in higher costs. At this end, policy intervention is advised to facilitate the process, because public institutions have the role and the capability to favor a better use of existing knowledge [69]. The suggestion is to implement policy tools to help private initiative facing the high transaction costs of ICT implementation jointly with end users. At this end, operational groups funded by the Rural Development Program (RDP) are a good example, bringing together different stakeholders with farmers. Similarly, the RDP's subsidies to investments on innovation implementation can be a powerful tool to directly finance investments on new platforms or indirectly by means of cross-compliance. Finally, ICT-development policies are advised to foster common metrics for ICT performance estimations. Often, ICT-developers deliver information on ICT capabilities which is not verified and comparable with other platforms. This again brings to unclear settings and risks to feed skepticisms to this new category of tools.

(2) Uncertainty-management policies would be needed to lower ambiguity on ICT reliability, speeding up the process of familiarity. Once new platforms are brought to the market, it would be ideal to offer consultancy for the implementation, long trials and demonstrative or formative events. Rather than a plug and play approach, these initiatives would allow end-users to better understand how information can be implemented and to gain experience on ICT reliability. Having hands on the platform, without necessarily implementing its information at DM's own expenses, would lower ambiguous perceptions and potentially foster the diffusion of ICTs. In addition, even after ICT adoption, DMs can be encouraged in starting to use the new platforms for informative purposes before attaching real decision making on it; this way they would experience ICT reliability without risking losses. As we modeled in this research, ICT-information implementation often implies moving from inefficient PPs with sure outcome, to efficient ICT-informed RP with uncertain outcomes. Here, if a DM is willing to bear such uncertainties to save water, he would be needing support in his virtuous choice. Accordingly, even after ambiguity is solved, uncertainty remains in the form of risk. Therefore, ex-post risk coping policies would be helpful to compensate losses at the WA's or farms' level when the ICT failed in its predictions.

(3) Agricultural and water policies instruments are suggested to be evaluated also with respect to their effects on risk perception to promote ex-ante risk management solutions to increase the sector's resilience. Between these, at the farm level, policies could favor investments in resource-efficient crops and irrigation systems; at the WA level, there could be favored: reservoirs, to face longer periods of scarcity; and investment in the irrigation network to allow efficient water allocation between districts. Policies for efficient water governance would be needed too. Here, the main aim would be to avoid excess-use of water by some farmers, which might cause production losses to others. Further, we have to consider that the share of risk estimated by the ICT is subjected to climate variability. This complicates ICT-informed decisions with CC, because every time the share of risk varies, the DM's expected utility from information implementation varies too. This issue will require DMs to take time to analyze case by case the uncertainty settings, before deciding. Therefore, policies for digitalization are suggested to account for such extra time and compensate adopters for their decision.

The above described policies call for two kinds of model applications: one is related with the estimation of potential ICT-benefits; the other with the assessment of impacts of AA in the process of ICT implementation. If research will be developed in these ways, it will provide the evidences needed by policy makers for effective support to the sector. On the one hand, assessments of potential ICT-benefits would allow DMs being less doubtful on the economic performances of ICT tools for irrigated



agriculture. These would help investments for ICT development and adoption, overall facilitating the transition from precautionary decisions based on experience to ICT-supported irrigation plans. Further, empirical assessments would allow to highlight restrictions for information usability which undermine ICT-benefits. This would support ICT-developers in targeting technologies to fit end users' informational needs. On the other hand, because in this transition AA has the potential to undermine information implementation, the impacts of DM's behavior must be assessed. Here, better insights on the behavioral perspective affecting ICT adoption would highlight the critical issues in the decision system to be targeted with ambiguity-management policies.

**Author Contributions:** The paper constitutes a collective effort of the four authors. Nevertheless, authors main contribution to each section of the research can be described as follows: conceptualization, F.C.; methodology, F.C.; investigation, F.C., F.G., M.R. and D.V.; writing—review, F.G. and D.V.; visualization, F.C., F.G., M.R. and D.V.; supervision, M.R. and D.V.; writing—original draft preparation, F.C. All authors have read and agreed to the published version of the manuscript.

**Funding:** This research received no external funding.

**Acknowledgments:** Previous versions of this paper are part of the doctoral dissertation of Francesco Cavazza: Cavazza, Francesco (2020), "The digital irrigated agriculture: advances on decision modelling to accompany the sector in exploiting new opportunities", Alma Mater Studiorum University of Bologna. DOI 10.6092/unibo/amsdottorato/9308.

**Conflicts of Interest:** The authors declare no conflict of interest.

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
