# Peer review of "Digital Irrigated Agriculture: Towards a Framework for Comprehensive Analysis of Decision Processes under Uncertainty"

_futureinternet, doi:10.3390/fi12110181_

Round 1

Reviewer 1 Report

The following paragraphs from the paper is practically the same as in thesis F. Cavazza The digital irrigated agriculture: advances on decision modelling to accompany the sector in exploiting new opportunities, 2020 on page 30.

However, the thesis is not among references.

Paragraph from the paper:

Besides attitudes toward uncertainty, the perception of uncertainty over a phenomenon is

extremely important in affecting decisions for ICT adoption (Nesheim et al. 2017). Further, in case of

information provision, also the perceived uncertainty over a new piece of information describing the

same phenomenon is found to be limiting information uptake (An-Vo et al. 2019). One strategy

proposed to solve this uncertainty is in the estimation of the forecast reliability and in the

incorporation of this information in the message itself (Kusunose and Mahmood 2016). This is

commonly done with probabilistic climate forecasts, but still DMs could doubt on the probability

estimation in itself. This explains one of the reasons why DM’s beliefs are needed to be accounted for

in decision analyses (Hardaker and Lien 2010). Nevertheless, in literature important challenges

remain in understanding the role of perceptions and attitudes (Jeuland et al. 2018) and most of the

studies in applied economics make relevant behavioral assumptions to overcome the issue

(Bobojonov et al. 2016). This gap in applied economics motivated us to seek for theories in the broader

economic literature to build the theoretical basis needed to develop research and better understand

decision processes of ICT implementation under uncertainty.

From thesis:

Besides attitudes toward uncertainty, the perception of uncertainty in itself is extremely important in affecting decisions for ICT adoption (Nesheim et al. 2017) and perceived uncertainty over forecasts reliability is found to be limiting information uptake (An-Vo et al. 2019). One strategy proposed to solve this uncertainty is in the estimation of the forecast reliability and in the incorporation of this information in the message itself (Kusunose and Mahmood 2016). This is commonly done with probabilistic climate forecasts, but still DMs could doubt on the probability estimation in itself. This explains one of the reasons why DM’s beliefs are needed to be accounted for in decision analyses (Hardaker and Lien 2010). Nevertheless, in literature important challenges remain in understanding the role of perceptions and attitudes (Jeuland et al. 2018) and most of the studies in applied economics make relevant behavioral assumptions to overcome the issue (Bobojonov et al. 2016). This gap in applied economics motivated us to seek for theories in the broader economic literature to understand the problem.

Page 1: add space before S: Abstract:Several

Please make the objective more concise, avoid tautology:

The novelty of this research is in the application of economic literature’s theories to provide the

108 theoretical basis needed to model the uncertainty settings around ICT adoption for irrigation

109 management.

Please correct double dot:

123 application is still predominantly pioneering..

Please correct error:

124 published on the topic (Error! Reference source not found.) and by the interest raised in the applied

Could adoption of technology be correlated with the number of papers (body of articles)?

This is confirmed by the growing body of articles

124 published on the topic (Error! Reference source not found.) and by the interest raised in the applied

125 economic literature (Jeuland et al. 2018; Martin 2016; Giupponi 2014).

Edit the information on source:

128 Source: own elaboration from data obtained with the search in Scopus (dated 04/04/2019) having

129 the following parameters: TITLE-ABS-KEY ((ICT OR DSS OR "climate

130 services”) AND agriculture AND ( "water management" OR irrigation))

Correct error:

156 Such findings are confirmed by several quantitative studies, out of which a selection of empirical

157 application for ICT benefits estimations is reported in Error! Reference source not found.. The table

Axiom is not clear, what is “world” meaning:

216 uncertainty (Hardaker and Lien 2010). The theory is defined on the basis of the following four axioms:

217 ï‚· A1 – Completeness: every state of the world involved in a decision can be completely

218 ranked.

Are states system states or governmental states in Axiom 2? ->

A2 – Transitivity: the property of transitivity holds for preferences 219 for alternative

220 states.

Please announce the following section in the text at the conclusions:

474 (1) ICT-development policies are needed to overcome issues of information usability

475 and boost ICT-potentials. The simple information provision is not sufficient to allow

476 its implementation because of local specificities in the end-user’s information

Author Response

Page 1: add space before S: Abstract:Several

--> done

Please make the objective more concise, avoid tautology:

--> I tried to be more concise by avoiding the tautology. But the objectives are still a bit long and complex due to the needs of the topic

Please correct double dot:

--> done

Please correct error:

124 published on the topic (Error! Reference source not found.) and by the interest raised in the applied

--> done

Could adoption of technology be correlated with the number of papers (body of articles)? This is confirmed by the growing body of articles

--> It can be, I corrected specifying and using conditional

124 published on the topic (Error! Reference source not found.) and by the interest raised in the applied

--> done

125 economic literature (Jeuland et al. 2018; Martin 2016; Giupponi 2014).

--> done

Edit the information on source:

--> done and corrected the description (there was an error in the day)

Correct error:

156 Such findings are confirmed by several quantitative studies, out of which a selection of empirical

157 application for ICT benefits estimations is reported in Error! Reference source not found.. The table

--> done

Axiom is not clear, what is “world” meaning:

--> it is a state of the world commonly used in literature to describe settings or physical states. I used italic to clarify

Are states system states or governmental states in Axiom 2? ->

--> see the answer before

Please announce the following section in the text at the conclusions:

--> done

Reviewer 2 Report

The summary includes the most important aspects of the study and provides enough information to start reading.
Keywords are adequate.
Objective: The research problem and the study objective are well defined.
the literature review is clearly presented. Despite this, this evaluator considers that the reason for the organization of the results in these sections specifically and not others should be explained to the reader.
The discussion of the data is correct and dynamic.
The conclusions are presented clearly.

Author Response

Thank you